# The Impact of Nutritional Status on Sexual Function in Male Kidney Transplant Recipients

**DOI:** 10.3390/medicina59020376

**Published:** 2023-02-15

**Authors:** Diana Sukackiene, Robertas Adomaitis, Marius Miglinas

**Affiliations:** Institute of Clinical Medicine, Faculty of Medicine, Vilnius University, 03101 Vilnius, Lithuania

**Keywords:** sexual function, erectile dysfunction, sexual desire, nutritional status, kidney transplantation, handgrip strength, muscle mass, body composition, bioelectrical impedance analysis

## Abstract

*Background and Objectives*: Sexual function and nutritional status assessment are relevant topics in chronic kidney disease patients. This study was designed to investigate whether bioelectrical impedance analysis-derived nutritional parameters, nutritional biomarkers, and handgrip strength influence sexual function and to analyze the changes in sexual function after kidney transplantation in men. *Materials and Methods*: Fifty-four men with kidney failure who had undergone replacement therapy entered the study. In addition, sexual function and nutritional status were evaluated before kidney transplantation and 12 months after. We used the International Index of Erectile Function, bioelectrical impedance analysis, three different malnutrition screening tools, handgrip strength, and anthropometric measurements. The demographic profiles and biochemical nutritional markers were collected. *Results*: Sexual inactivity was associated with a lower phase angle (6 (1) vs. 7 (1), *p* < 0.05) and a higher fat mass index (7 (5) vs. 3 (4), *p* < 0.05). Normal erectile function before kidney transplantation was significantly related to higher fat-free mass (67 (11) vs. 74 (7), *p* < 0.05) and lean mass (64 (10) vs. 70 (7), *p* < 0.05). The improvement in erectile function after kidney transplantation was nonsignificant (44% vs. 33%, *p* = 0.57). Only a weak association between muscle mass and sexual desire 12 months after kidney transplantation was found (rS = 0.36, *p* = 0.02). Further, linear regression revealed that higher muscle mass could predict better sexual desire after kidney transplantation following adjustment for age, estimated glomerular filtration rate, and diabetes mellitus. *Conclusions*: Kidney transplantation did not significantly improve erectile dysfunction in our study population. Sexual desire and intercourse satisfaction are the most affected domains in patients with chronic kidney disease. Higher muscle mass predicts higher sexual desire after kidney transplantation. Higher levels of fat-free and lean mass are associated with normal erectile function before kidney transplantation.

## 1. Introduction

Chronic kidney disease (CKD) is a leading public health problem worldwide, with a continuously rising incidence during the past decades, affecting quality of life despite the increase in life expectancy in these patients [1]. In addition, sexual dysfunction is highly prevalent in the CKD population. In total, 50% of kidney failure patients without replacement therapy (KRT) and 75% of kidney failure patients with replacement therapy (KFRT) report erectile dysfunction (ED), and these estimates are higher than those in the general population [2,3]. Further, disorders of sexual function and CKD share common pathophysiological pathways—vascular or hormonal abnormalities—and are the hallmarks of cardiovascular disease, hypertension, and diabetes mellitus (DM) [1].

Poor nutritional status due to protein-energy wasting (PEW), malnutrition, and sarcopenia are frequent complications associated with increased mortality in patients with CKD [4]. However, kidney transplantation (KT), championed as the best treatment for CKD, can also contribute to malnutrition and muscle weakness [5]. To date, the effects of KT on ED are very contradictory, ranging from recovery of potency to minimal effects on ED following KT [6,7].

Currently, a limited number of studies have investigated the impact of nutritional status on sexual function in KT recipients. Therefore, the primary endpoint of the study was whether any nutritional parameters influence any aspect of sexual function (sexual activity, erectile function, sexual desire, orgasmic function, intercourse satisfaction, or overall satisfaction) before and after KT. The secondary endpoint was the impact of KT on specific domains of sexual function.

## 2. Subjects and Methods

### 2.1. Study Design and Patients

This prospective observational study was carried out between January 2018 and March 2020 in a tertiary referral university hospital. The aim of this study was to evaluate male sexual function and investigate its associations with anthropometric measurements, handgrip strength, biochemical markers, and bioelectrical impedance analysis (BIA)-derived nutritional parameters at baseline (before KT) and 12 months posttransplant.

The inclusion criteria were as follows: (1) male sex; (2) age greater than or equal to 18 years; (3) KFRT (on hemodialysis (HD) or peritoneal dialysis (PD)) for more than 3 months; (4) being a kidney transplant recipient; and (5) signing an informed consent form. The exclusion criteria were as follows: limbless patients, patients with pacemakers, and patients who refused to participate in the study. The flowchart of patient selection is shown in Figure 1.

### 2.2. Ethics

The study was approved by the regional research ethics committee and complies with the principles of the Helsinki Declaration. Written informed consent was obtained from all patients before enrolment in the study.

### 2.3. Laboratory Data

The blood samples for the assessment of albumin and prealbumin were collected twice during the study: at baseline (before KT) and 12 months later, after a 12-h overnight fast. BD Vacutainer SST-II Advance Serum Separator Tubes (BD Diagnostics, UK) were used. Samples were centrifuged at the local clinical chemistry laboratory (3230 RCF, ambient temperature, 7 min) and analyzed the same day using standard automated methods.

### 2.4. Assessment of International Index of Erectile Function

The International Index of Erectile Function (IIEF) is an extensively used, self-report diagnostic tool for the evaluation of male sexual function and ED severity. The IIEF is a 15-item tool that examines five domains of male sexual function: erectile function (EF), orgasmic function (OF), sexual desire (SD), intercourse satisfaction (IS), and overall satisfaction (OS) [8]. Each question is scored on a 5-point scale. The total score range is 4–75, but the scales for each domain vary. A higher score indicates better sexual functioning. The severity of ED was based on the Cappelleri criteria: no ED (EF score 26 to 30), mild ED (EF score 22 to 25), mild to moderate ED (EF score 17 to 21), moderate ED (EF score 11 to 16), and severe ED (EF score 6 to 10) [9]. The questionnaire, translated into the native local language, was administered by a nephrologist before and at least 12 months after successful KT.

### 2.5. Evaluation of Nutritional Status

#### 2.5.1. Anthropometric Data

The anthropometric measurements for all participants were performed by one trained nephrologist. Height (cm) and weight (kg) were measured using an automatic scale with a sensitivity of 0.1 cm and a resolution of 0.1 kg (Seca, model 704, Hamburg, Germany). BMI was calculated as a ratio between weight and height in metres squared (kg/m^2^). The waist circumference (cm) was examined 2.5 cm above the umbilicus using a measuring tape.

#### 2.5.2. Handgrip Strength Assessment

Handgrip strength (HGS) is an important tool for representing overall muscle strength and diagnosing sarcopenia. HGS in our study was measured using a Saehan hydraulic hand dynamometer (Saehan, model SH5002, Changwon, Republic of Korea) with a scale of strength up to 100 kg. HGS was evaluated on the nonfistula dialysis arm or, if there was no fistula or PD patient, on the dominant arm. Three measurements were performed with an interval of 5 s between measurements, and the highest value was used for analysis.

#### 2.5.3. Bioelectrical Impedance Analysis

Bioelectrical impedance analysis is a technology that is widely used for the assessment of body composition. The purpose of BIA is to measure the electric impedance of an electric current passing through the body. We performed measurements by using an InBody S10 portable body bioimpedance spectroscopy device (Biospace, Seoul, Republic of Korea). The analysis was conducted with patients in a supine position 30 min after HD or when patients had an empty abdomen in PD patients, and all recommendations from the European Society for Clinical Nutrition and Metabolism (ESPEN) and the manufacturers were followed [10]. From the determined impedance, a number of BIA parameters were evaluated. Fat mass (FM) and fat-free mass (FFM) estimations are considered the main parameters in body composition assessment techniques. FFM is everything that is not body fat and decreases with age and chronic diseases. All cells that have an effect on metabolism compose body cell mass (BCM). Low BCM values are associated with malnutrition, dehydration, and cachexia. Phase angle (PhA) is one of the best markers of cell membrane function, characterizing training and nutritional status. Low PhA and low muscle mass (MM) indicated poor training status and poor nutritional status [11].

BIA-derived parameter indices (fat mass index (FMI), fat free mass index (FFMI), muscle mass index (MMI), lean mass index (LMI), and body cell mass index (BCMI)) were calculated by dividing each type of tissue weight in kilograms by height^2^ (kg/m^2^).

#### 2.5.4. Nutrition-Related Questionnaires

Nutritional risk screening tools are very helpful in daily clinical practice to detect potential or existing malnutrition. Currently, more than 33 different nutritional risk screening tools are available [12]. We chose the Subjective Global Assessment Scale (SGA), Malnutrition Inflammation Score (MIS), and Geriatric Nutritional Risk Index (GNRI), which have been previously reported to be applicable tools in CKD populations [13]. SGA and MIS were administered through face-to-face interviews. The GNRI does not require patient interviewing, and the scores were calculated from the serum albumin level and the ratio between ideal and actual body weight.

The Subjective Global Assessment Scale (SGA) is a validated tool to assess nutritional status and is recommended by the American Society of Parenteral and Enteral Nutrition (ASPEN). The SGA focuses on features of the medical history and physical examination and scores patients on a scale ranging from A-well nourished, B-mild to moderately malnourished, to C-severely malnourished [14].

For the MIS, the cut-off proposed by Yamada et al. was used to classify nutritional status: 0 to 5 well-nourished, 6 to 10 mild PEW, and ≥11 moderate to severe PEW [15].

### 2.6. Statistical Analysis

The results are expressed as the mean ± standard deviation (SD) or the median with interquartile range (IQR), and categorical variables are expressed as percentages. Data were tested for normal distribution by Shapiro–Wilk statistics. To check the equality of two groups, an F test and, if appropriate, a paired Student’s t test for normally distributed data or a paired Wilcoxon signed rank test for nonparametric variables were applied, and for categorical variables, a chi-square test was used. For correlations between nutritional parameters and sexual function (EF, OF, SD, IS, OS), Spearman’s rank test was performed.

Linear regression analysis was used to evaluate the association of muscle mass (adjusted for age, DM, and estimated glomerular filtration rate (GFR)) with sexual desire 12 months after KT.

A value of *p* lower than 0.05 was considered significant. All analyses were performed using R Commander (Rcmdr) version 3.3.2.

## 3. Results

### 3.1. Patient Characteristics

Fifty-two male kidney transplant recipients were enrolled. Baseline characteristics are presented in Table 1.

The underlying causes of kidney disease included glomerulonephritis (60%), polycystic kidney disease (12%), inherited kidney diseases (13%), diabetic nephropathy (8%), amyloidosis (2%), pyelonephritis (2%), and kidney stones (4%). A total of 50 out of 52 enrolled patients (96 %) had primary or secondary hypertension. A total of 46% of patients received four or more antihypertensive agents. Only 8% were taking monotherapy, 27% were taking dual therapy, and 19 % triple therapy. Among all antihypertensive classes, the most common were calcium channel blockers (67%) and beta blockers (65%). Other antihypertensive agents were used more rarely: centrally acting (48%), angiotensin receptor blockers (ARB) (44%), diuretics (33%), alpha blockers (29%), and angiotensin-converting enzyme (ACE) inhibitors (13%).

Identical immunosuppressive treatment with a combination of a calcineurin inhibitor, methylprednisolone, and mofetil mycophenolate was prescribed to all participants. The mean cold ischaemia time was 14 h 48 min ± 7 h, and in 34 cases (65%), delayed graft function without acute transplant rejection was observed.

### 3.2. Sexual Activity and Nutritional Status in Kidney Transplant Recipients before Kidney Transplantation

We compared nutrition-related variables between sexually active (*n* = 41) and inactive (*n* = 11) KT recipients before KT (Table 2).

The majority of sexually active men (93%) and only one-third (36%) of sexually inactive men had a sexual partner. Obesity was prevalent in 46% of sexually inactive men and in 15% of sexually active men. In addition, sexually active men had significantly higher BIA-derived PhA but lower body fat percent and fat mass index.

### 3.3. Nutritional Parameters and Erectile Function before and after Kidney Transplantation

Those who reported no sexual activity before KT remained sexually inactive after KT. Therefore, further data analysis was performed only with sexually active men.

The prevalence of ED before KT was 44% (*n* = 18), and that following KT dropped to 33% (*n* = 13) (*p* = 0.57). All patients (*n* = 3) with DM had ED before and after KT. The results of ED are summarized in Table 3.

Sexual desire and intercourse satisfaction were the two most affected domains (*p* < 0.05). No significant differences were found among any of the five domains of the IIEF-15 before and 12 months after KT (EF *p* = 0.57, OF *p* = 0.15, SD *p* = 0.33, IS *p* = 0.15, OS *p* = 0.54) (Figure 2).

The BIA-derived nutritional parameters in men with ED (score < 26) compared to those with normal erectile function (score > 26) are shown in Table 4. Significant pretransplant differences were found in fat-free mass (67 ± 11 vs. 74 ± 7, *p* < 0.05) and soft lean mass (64 ± 10 vs. 70 ± 7, *p* < 0.05). However, no differences were identified 12 months posttransplant.

We also analysed whether BIA-derived nutritional parameters could predict sexual function (erectile function, orgasm, sexual desire, intercourse satisfaction, overall sexual satisfaction) before and after KT. A weak association between muscle mass and sexual desire 12 months after KT was identified (Figure 3). Linear regression (Table 5) revealed that if two patients differ by 10 kg of muscle mass, we expect that the one with more muscle mass will have a SD +1 point higher, an average, after accounting for age, DM, eGFR.

## 4. Discussion

Sexual dysfunction is a common feature that reduces CKD patients’ quality of life and is often underestimated by clinicians [16]. Successful KT is a preferred renal replacement therapy for KFRT patients, providing better health-related quality of life [17]. The present study showed that kidney transplantation may not improve erectile function after one year of follow-up [18,19]. Impaired sexual desire was associated with lower muscle mass in male kidney transplant recipients.

In this population, ED has a multifactorial etiology, including abnormalities in the hypothalamic-pituitary-gonadal axis, autonomic nervous system disturbances, endothelial dysfunction, anaemia, secondary hyperparathyroidism, medication effects, and neurological and psychological derangements [20,21]. Disturbances on sex steroid synthesis, secretion, and metabolism (leading to primary hypogonadism and dysfunction of the hypothalamic-pituitary axis) occur with a degree of severity proportionate to the reduction in glomerular filtration rate [22]. High levels of follicle stimulating hormone (FSH), luteinizing hormone (LH), prolactin, and low testosterone levels are reported in uremic men. Kidney transplantation has been demonstrated to reverse the impaired endocrine status in the majority of recipients. The restoration of kidney function improves prolactin clearance, which leads to normalization of LH and testosterone levels [6,23].

Furthermore, ED and CKD are associated with diseases causing endothelial impairment, such as hypertension, diabetes, dyslipidemia, obesity, and metabolic syndrome [24]. Endothelial dysfunction plays an important role in ED, as nitric oxide (NO) production is decreased in poorly functioning endothelial cells. Therefore, CKD directly (via impact on endothelial function) and indirectly (via associated metabolic conditions) influence ED [25].

Sexual inactivity or absence of sexual intercourse is usually considered an exclusion criterion [19]. However, we suppose that sexual inactivity is one of the important puzzle details in evaluating sexual function in CKD patients. Changes in body shape and image (catheter, fistula) also contribute to weaker sexual desire and dysfunction [1]. In accordance with Goncalves et al. [26], we showed that 11 male kidney transplant recipients reported no sexual activity in the previous 4 weeks (21%). The European Male aging Study indicated adiposity as a risk factor for sexual dysfunction [27]. Additionally, it has been reported that testosterone and sex-hormone binding globulin levels are lower in men with obesity and inversely correlate with the degree of obesity [28,29].This condition triggers a vicious cycle in which low testosterone contributes to maintaining a high body weight and accumulation of intra-abdominal fat, leading to a greater degree of testosterone deficiency [30]. In our report, obesity, a lower phase angle, and a higher fat mass index were associated with sexual inactivity.

A higher phase angle indicates better nutritional status [31]. In fact, nutritional issues (malnutrition, protein energy wasting, and sarcopenia) are also highly relevant in all phases of CKD, even in KT [32]. Posttransplant immunosuppressive therapy, e.g., glucocorticoids, calcineurin inhibitors, and mofetil mycophenolate, affects body composition by causing the accumulation of fat mass and the loss of muscle mass [33]. We found that the greater the muscle mass, the better the sexual desire. In addition, sexually active men with ED before KT had significantly lower fat-free mass and lean mass. It is likely that measures that result in gaining muscle mass would be helpful in increasing sexual desire after KT.

Sexual desire and intercourse satisfaction were the most affected domains in men after KT. Raggi et al. found that both the desire and the frequency of sexual activity decreased after transplantation [34]. Al Khallaf summarized that orgasmic function is the only sexual function spared among male dialysis patients and kidney transplant recipients [19]. Several studies reported improvements in erectile parameters after KT [35,36,37]. However, our results are in agreement with the authors, who showed that there was no significant difference in erectile function between uremic patients who receive kidney transplant and dialysis [18,19]. The main change in erectile dysfunction was observed in patients with mild dysfunction to no dysfunction, while the rest of the ED severity groups remained unaltered before and after KT.

In addition, we found no significant difference when comparing the mean values of the total IIEF score, erectile function, orgasmic function, intercourse satisfaction, or overall satisfaction score before and 12 months following kidney transplantation. It is worth mentioning that all patients received transplants in which an end-to-side vascular anastomosis was created between the renal artery of the graft and the external or common iliac artery of the recipient. Therefore, vascular anastomosis did not affect erectile function in our cohort.

Due to the small number of living donations in our transplantology center, we could not compare sexual function between living and deceased kidney donor recipients. Interestingly, Branco et al. could only observe a higher level of intercourse satisfaction among living donor recipients [38].

Some limitations need to be considered. First, we did not measure testosterone levels, which are meaningful in sexual disorders. Furthermore, the study had a relatively small sample size, which may have reduced the statistical power. Another limitation is that the BIA measurement used in the present study has lower accuracy in monitoring body composition than dual-energy X-ray absorptiometry. However, BIA is an easily applicable bedside method that is widely used in daily practice. Finally, quality of life and physical activity were not evaluated, which could be useful in interpreting the results.

Despite these limitations, the present study provides valuable information as the first exploratory study to identify how the specific changes in body composition are related to male sexual function after kidney transplantation.

## 5. Conclusions/Future Directions

This study confirmed that ED may persist after successful kidney transplantation. Sexual desire is one of the most impaired domains in male KT recipients and is associated with lower muscle mass. In addition, the promotion of physical activity and a healthy diet together with the management of immunosuppressive therapy may have beneficial effects in terms of the improvement of sexual dysfunction in this population before and after KT. Further, sexual function assessment should be considered an essential part of the clinical assessment of kidney transplant recipients. Early diagnosis of sexual dysfunction and early referral to specialists leads to improvements in mental health and quality of life.

Additionally, large-scale prospective studies examining the impact of nutritional parameters and their relationship with sexual function in KT patients are needed to substantiate the findings of the present study.

## Figures and Tables

**Figure 1 medicina-59-00376-f001:**
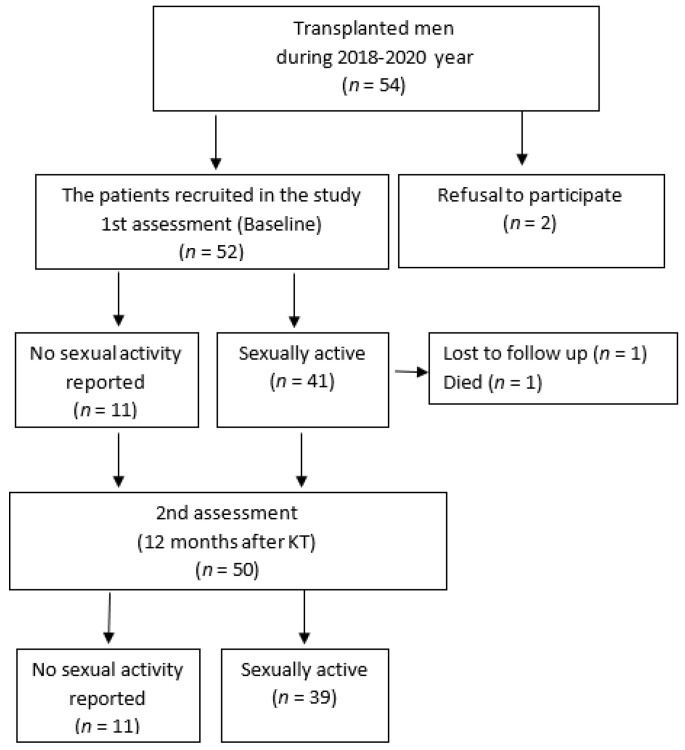
Study flowchart.

**Figure 2 medicina-59-00376-f002:**
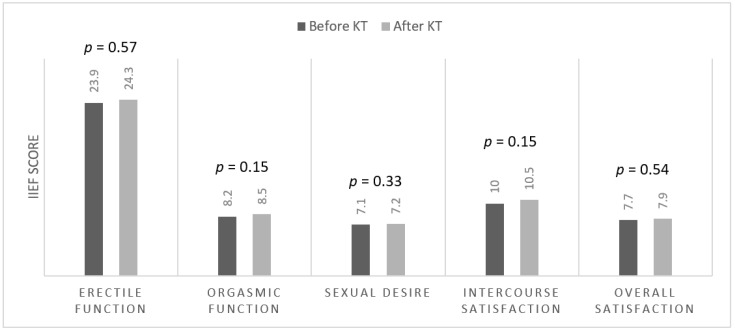
IIEF-15 score before and after KT.

**Figure 3 medicina-59-00376-f003:**
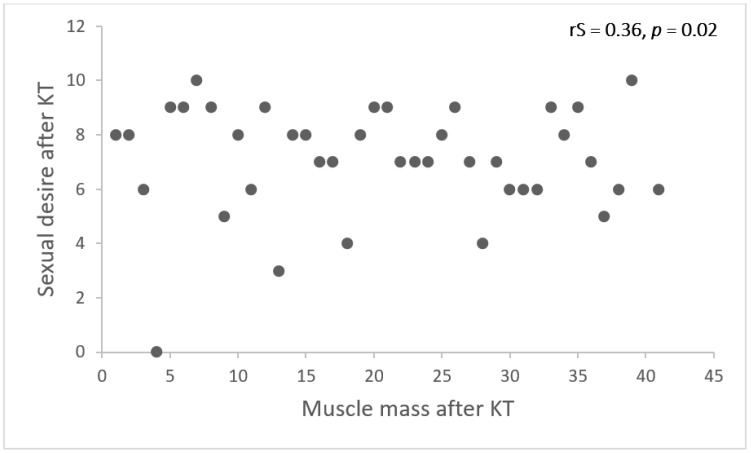
Correlation between muscle mass and sexual desire 12 months after. KT.rS—Spearman correlation coefficient.

**Table 1 medicina-59-00376-t001:** Pretransplant characteristics.

Variable	*n* = 52
Age, year	40.8 (11)
Residual kidney function, yes	31 (60)
SGA	A 30 (58) B 22 (42)
GNRI score	114 (10)
MIS score	4 (2)
Dialysis modality	HD 42 (81) PD 10 (19)
Dialysis vintage, months	19 (20)
Having a partner, yes	42 (81)
Comorbidities Diabetes Hypertension Cardiovascular disease Dyslipidemia Neurological disorders	4 (8) 50 (96) 9 (17) 33 (63) 3 (6)

Data expressed as mean/ median (SD/IQR) or percent (number of patients); SGA—Subjective Global Assessment, GNRI—Geriatric Nutritional Risk Index, MIS—Malnutrition Inflammation Score.

**Table 2 medicina-59-00376-t002:** Comparison of nutrition related variables between sexually active and inactive recipients before kidney transplantation.

Variable	No Sexual Activity (*n* = 11)	Sexually Active (*n* = 41)	*p* Value
Dialysis vintage	16 (17)	19 (21)	0.76
Age, year	44 (13)	40 (10)	0.20
Weight, kg	91 (25)	84 (15)	0.35
BMI, kg/m^2^	28 (6)	25 (4)	0.05
Muscle mass, kg	39 (9)	40 (6)	0.67
Muscle mass index kg/m^2^	12 (2)	12 (1)	0.90
Lean mass, kg	66 (14)	67 (9)	0.84
Handgrip strength, kg	44 (8)	48 (11)	0.26
Phase angle, °	6 (1)	7 (1)	0.04
Lean tissue index, kg/m^2^	20 (3)	20 (2)	0.76
Fat mass, kg	22 (15)	11 (12)	0.07
Fat mass index, kg/m^2^	7 (5)	3 (4)	0.04
Body fat (%)	22 (15)	13 (12)	0.05
Fat free mass, kg	70 (16)	71 (9)	0.85
Fat free mass index, kg/m^2^	22 (4)	22 (2)	0.73
Body cell mass, kg	45 (10)	46 (6)	0.75
Body cell mass index, kg/m	14 (2)	14 (2)	0.89
Waist circumference, cm	105 (21)	94 (12)	0.13
MIS score	4 (2)	5 (2)	0.83
GNRI score	118 (12)	114 (10)	0.24
Prealbumin, mg/dL	43 (8)	42 (8)	0.81
Albumin, g/L	45 (3)	45 (4)	0.81

Data expressed as mean/median (SD/IQR) or percent (number of patients). BMI—body mass index, GNRI—Geriatric Nutritional Risk Index, MIS—Malnutrition Inflammation Score.

**Table 3 medicina-59-00376-t003:** ED rate before and after KT.

Dysfunction Rate.					
	Severe, *n* (%)	Moderate, *n* (%)	Mild to Moderate, *n* (%)	Mild, *n* (%)	No Dysfunction, *n* (%)
Before KT (*n* = 41)	3 (7%)	2 (5%)	6 (15%)	7 (17%)	23 (56%)
After KT (*n* = 39)	3 (8%)	2 (5%)	5 (13%)	3 (8%)	26 (66%)
*p*-value	0.94	0.95	0.81	0.20	0.33

**Table 4 medicina-59-00376-t004:** Comparison of nutrition related variables with regards to ED presence before and after KT.

	Before KT (*n* = 41)			12 Months after KT (*n* = 39)		
	ED (*n* = 18)	No ED (*n* = 23)	*p* Value	ED (*n* = 13)	No ED (*n* = 26)	*p* Value
Age, year	40 (12)	39 (9)	0.79	42 (13)	41 (9)	0.76
Dialysis vintage, months	23 (21)	18 (20)	0.62	-	-	-
Delayed graft function, yes (%)	-	-	-	7 (54%)	16 (61%)	0.64
BMI, kg/m^2^	25 (4)	26 (4)	0.28	25 (8)	25 (4)	0.83
Waist circumference, cm	91 (16)	91(16)	0.46	97 (11)	98 (12)	0.67
Handgrip strength, kg	50 (10)	46 (12)	0.19	44 (11)	53 (15)	0.13
Muscle mass, kg	38 (7)	42 (4)	0.05	36 (4)	38 (5)	0.17
Muscle mass index, kg/m^2^	12 (2)	12 (1)	0.25	11 (1)	12 (1)	0.59
Fat mass, kg	10 (9)	12 (15)	0.41	13 (9)	15 (12)	0.43
Body fat %	14 (9)	14 (8)	0.78	18 (8)	19 (9)	0.81
Fat free mass, kg	67 (11)	74 (7)	0.04	65 (7)	68 (8)	0.19
Fat free mass index, kg/m^2^	21 (3)	22 (2)	0.24	20 (2)	21 (2)	0.66
Fat mass index, kg/m^2^	3 (3)	4 (4)	0.58	5 (4)	4 (3)	0.98
Body cell mass, kg	44 (7)	48 (4)	0.05	42 (4)	44 (5)	0.17
Body cell mass index, kg/m^2^	14 (2)	14 (2)	0.27	13 (1)	13 (1)	0.62
PhaseA, °	7 (1)	7 (1)	0.77	6 (1)	6 (1)	0.26
Lean mass, kg	64 (10)	70 (7)	0.03	61 (6)	65 (8)	0.19
Lean tissue index, kg/m^2^	20 (2)	21 (2)	0.24	18 (2)	19 (2)	0.43
Albumin, g/l	46 (5)	45 (4)	0.76	44 (3)	45 (3)	0.07
Prealbumin, mg/dL	43 (8)	43 (8)	0.83	33 (8)	37 (8)	0.25
GNRI score	112 (10)	115 (9)	0.38	112 (6)	115 (9)	0.21
MIS score	5 (2)	5 (2)	0.83	1(1)	1(2)	0.61

Data expressed as mean/median (SD/IQR) or number of patients (percent); BMI—body mass index, GNRI—Geriatric Nutritional Risk Index, MIS—Malnutrition Inflammation Score.

**Table 5 medicina-59-00376-t005:** Linear regression analysis. Muscle mass as determinant of sexual desire —following adjustments for age, diabetes mellitus and eGFR 12 months after KT.

	Sexual Desire		
	Beta	SE	*p*-Value
Muscle mass, kg	0.102	0.05	0.04

eGFR—estimated glomerular filtration rate.

## Data Availability

The data presented in this study are available on request from the corresponding author. The data are not publicly available due to ethical restriction.

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
