# Peer review of "The Impact of Nutritional Status on Sexual Function in Male Kidney Transplant Recipients"

_medicina, 2023, doi:10.3390/medicina59020376_

Round 1
Reviewer 1 Report
Sukackiene et al. assessed the impact of nutritional status and kidney transplantation on sexual male function in kidney transplant recipients. Overall the study targets an interesting topic and is well presented and discussed. In this study KT did not improve sexual function. Interestingly, there were correlations between nutritional status and post transplant sexual function.
Major comments:
- The overall n, especially for subgroup analysis is relatively low. Ideally, additional data should be collected or subgroups should be merged for sufficient sample size
- It would be interesting to see results of similar research in female patients
- The authors discuss the impact of obesety on sex-hormone levels. However, references on sex-hormone levels in CKD and dialysis patients are not discussed and should be added.
Minor comments:
- Fig 2: Single datapoints additionally to the box plots would be interesting to see.
Author Response
Dear Reviewer,
Thank you for your insights and comments, and most of all for your time spent in evaluating our manuscript. Please find our response in the attached word file.

Reviewer 2 Report
This is small but properly conducted observational study assessing erectile dysfunction before and after kidney transplantation. Authors focused on body composition parameters, namely fat / fat-free mass, nutritional markers, muscle strength (handgrip).
Methodology is typical , population description satisfactory.
The main finding if the stydy is that patients who reported no sexual activity before KT remained sexually inactive after KT. Erectile dysfunction was also associated with less fat free mas.
The only missing point is vascular health data. Best if endothelial markers will be given. Authors must report blood pressure , antihypertensive medication distribution. Erectile dysfunction is occasional side effect of BP drugs like thiazide diuretics, loop diuretics, and beta-blockers, all of which can decrease blood flow to the penis and make it difficult to get an erection.
Most probably active physically mens (already before KTx) has higher muscle mass , less fat and functional endothelium. Endothelial function should be mentioned in discussion section.
Author Response

(The authors gave the same response as above.)

Reviewer 3 Report
REVIEW
The Impact of Nutritional Status on Sexual Function in Male
Kidney Transplant Recipients
The authors in the current article describe the Impact of Nutritional Status on Sexual Function in Male Kidney Transplant Recipients. As far as I am concerned the problem is very important, since more and more patients wait for organ transplantation. In the mentioned group of patients there are also young patients, young man who would like to have family. Expanding the knowledge by properly conducted researches is extremely important and crucial.
I find the current article very atracive, clearly written with the proper conclusion.
„Kidney transplantation did not significantly improve erectile dysfunction in our study population. Sexual desire and intercourse satisfaction are the most affected domains in chronic kidney disease patients. Higher muscle mass predicts higher sexual desire after kidney transplantation. Higher fat-free and lean mass are associated with normal erectile function before kidney transplantation.”
I would only suggest to add more recent citations.
Author Response
Dear. Reviewer,
Thank you for your insights and comments, and most of all for your time spent in evaluating our manuscript. Please find our response below:
I would only suggest to add more recent citations.
Thank you for this suggestion. We made changes and added more recent references in the manuscript.